# Achieving Tunable Mechanoluminescence in CaZnOS:Tb^3+^, Sm^3+^ for Multicolor Stress Sensing

**DOI:** 10.3390/nano14151279

**Published:** 2024-07-30

**Authors:** Wenqi Wang, Zihui Li, Ziying Wang, Zhizhi Xiang, Zhenbin Wang, Sixia Li, Mingjin Zhang, Weisheng Liu

**Affiliations:** 1School of Chemistry and Chemical Engineering, Qinghai Normal University, Xining 810008, China; 2Qinghai Key Laboratory of Advanced Technology and Application of Environmental Functional Materials, Xining 810016, China; 3Key Laboratory of Nonferrous Metals Chemistry and Resources Utilization of Gansu Province and State Key Laboratory of Applied Organic Chemistry, College of Chemistry and Chemical Engineering, Lanzhou University, Lanzhou 730000, China

**Keywords:** mechanoluminescent, multicolor, CaZnOS, co-doped, Tb^3+^, Sm^3+^

## Abstract

Mechanoluminescent (ML) materials can exhibit visible-to-near-infrared mechanoluminescence when responding to the fracture or deformation of a solid under mechanical stimulation. Transforming mechanical energy into light demonstrates promising applications in terms of visual mechanical sensing. In this work, we synthesized the phosphor CaZnOS:Tb^3+^, Sm^3+^, which exhibited intense and tunable multicolor mechanoluminescence without pre-irradiation. Intense green ML materials were obtained by doping Tb^3+^ with different concentrations. Tunable multicolor mechanoluminescence (such as green, yellow-green, and orange-red) could be realized by combining green emission (about 542 nm), attributed to Tb^3+^, and red emission (about 600 nm) generated from the Sm^3+^ in the CaZnOS substrate. The tunable multicolor ML materials CaZnOS:Tb^3+^, Sm^3+^ exhibited intense luminance and recoverable mechanoluminescence when responding to mechanical stimulation. Benefiting from the excellent ML performance and multicolor tunability in CaZnOS:Tb^3+^, Sm^3+^, we mixed the phosphor with PDMS and a curing agent to explore its practical application. An application for visual mechanical sensing was designed for handwriting identification. By taking a time-lapsed shot while writing, we easily obtained images of the writer’s handwriting. The images of the ML intensity were acquired by using specific software to transform the shooting data. We could easily distinguish people’s handwriting through analyzing the different ML performances.

## 1. Introduction

Mechanoluminescent (ML) materials possess the ability to emit visible-to-near-infrared mechanoluminescence by responding to mechanical stimulation, and are widely used in modern technologies such as visual mechanical sensing, optical sensors, and artificial skin [1,2,3,4]. Composed of inorganic crystals or organic complexes, they are promising for applications in visual mechanical sensing [5]. There are plentiful mechanoluminescent materials that have been reported since they were discovered [6,7,8,9,10], for example, the visible-to-near-infrared mechanoluminescence in SrZn_2_S_2_O and SrZnSO, the doping of lanthanide ions (e.g., Tb^3+^, Eu^3+^, Pr^3+^, Sm^3+^, Er^3+^, Dy^3+^, Ho^3+^, Nd^3+^, Tm^3+^, and Yb^3+^) in CaZnOS excited different mechanoluminescence, and the tuning of mechanoluminescence in lanthanide(III) and manganese(II) co-doped CaZnOS crystals [11,12,13]. However, the potential development of ML phosphors was greatly limited due to the limitations of the color. Therefore, the development of multiple tunable colors in a single substrate should be one of the most urgent tasks in ML material research [14,15].

Due to the wide application of ML materials, researchers have been working on the development of novel ML materials and have discovered a great number of compounds with ML properties [16,17,18]. Peng et al. reported the incorporation of abundant lanthanide ions into CaZnOS crystals, achieving tunable mechanoluminescence spanning the full spectrum from violet to near-infrared [11]. Wang et al. demonstrated in their excellent work that CaZnOS:Er/Mn, a multicolor and multimode mechanical luminescent material, showed promising application prospects in spectral regulation and information coding [12]. Xie et al. reported the incorporation of lanthanide ions or transition metals into the mixed-anion compounds SrZn_2_S_2_O and SrZnSO, creating visible-to-near-infrared mechanoluminescence [13]. Enormous potential has been shown by multicolor ML materials in practical applications [19,20,21]. Therefore, great significance should be attached to developing new multicolor ML materials.

Most of the luminescent materials that have been reported so far exhibit hardly any recoverable emissions and achieve multiple tunable colors in a single substrate when mechanically excited. Here, we developed a meaningful strategy that could achieve a tunable multicolor mechanoluminescence in a single substrate of CaZnOS co-doped with Tb^3+^ and Sm^3+^ ions. CaZnOS was chosen as the host material in our study owing to its convenience in terms of impurity doping and its band gap for transition. A series of CaZnOS:Tb^3+^, Sm^3+^ phosphors were investigated by characterizations and performance tests in detail, such as X-ray diffraction (XRD), scanning electron microscopy (SEM), energy-dispersive spectroscopy (EDS), transmission electron microscopy (TEM), X-ray photoelectron spectroscopy (XPS), the diffuse reflectance spectrum (DRS), the excitation spectrum, and the emission spectrum. We demonstrated the visual mechanical sensing of handwriting identification with the use of CaZnOS:Tb^3+^, Sm^3+^/PDMS composite elastomer. From the pictures, it was clear to see that different ML performances were exhibited with different writing stresses. This type of method possesses more potential applications in visual mechanical sensing, which can help with the synthesis of novel materials and expand the promising applications of ML phosphors into more domains.

## 2. Experiments

### 2.1. Sample Preparation

The phosphor samples of Ca1-x-yZnOS:xTb^3+^, ySm^3+^ (x = 0.01, 0.02, 0.03, 0.04, and 0.05, where x = 0.03, y = 0.0005, 0.001, 0.002, 0.005, and 0.01) were synthesized using the conventional high-temperature solid-state method. Raw materials of the sample were CaCO_3_ (A.R.), ZnS (A.R.), Tb_4_O_7_ (A.R.), Sm_2_O_3_ (A.R.), and Li_2_CO_3_ (A.R.). The corresponding starting materials were weighed by stoichiometric amounts and placed into an agate mortar. Subsequently, an adequate quantity of ethanol was incorporated with the starting materials, mixed homogeneously, and ground thoroughly in the agate mortar for 25 min. Next, placed into alumina crucibles, the mixture was sintered at 1100 °C for 3 h under an argon gas atmosphere (argon gas at 40 mL/min) in an electric tube furnace. For subsequent use, the samples were cooled to room temperature and ground again.

### 2.2. Fabrication of CaZnOS:Tb^3+^,Sm^3+^/PDMS Composite

Polydimethylsiloxane (PDMS) served as an elastic matrix, which offered internal stress to the ML phosphor [5,22]. First of all, the phosphor was ground into more uniform particles in an agate mortar. Second, 1 g of PDMS (Sylgard 184, Shanghai Deji Trading Co., Ltd., Shanghai, China) and 0.1 g of curing agent were thoroughly mixed in the mold. Third, the 1.1 g of phosphor was weighed and evenly mixed with the prepared mixture. To dry and solidify the mixture, it was placed in a drying oven at 65 °C for 30 min. Finally, the CaZnOS:Tb^3+^, Sm^3+^/PDMS composite was acquired.

### 2.3. Measurements and Characterization

Rigaku D/Max-2400 X-ray (Rigaku, Tokyo, Japan) diffraction with copper target Kα was used to analyze the crystal structure of the materials, covering a detection range from 10° to 90° at a speed of 20° per minute. Based on the Rietveld refinement pattern of the Ca_0.96_ZnOS:0.03Tb^3+^, 0.01Sm^3+^ sample, the crystal structure of the phosphor sample was acquired. Transmission electron microscopy images were captured with an FEI Tecnai F30 instrument (Philips-FEI, Amsterdam, The Netherlands). The morphology and elemental mapping of typical samples were characterized using a Zeiss ZEISS-6035 scanning electron microscope (CarlZeiss, Oberkochen, Germany). X-ray photoelectron spectroscopy analysis was conducted using a Thermo ESCALAB 250 Xi system from Thermo Fisher Scientific Inc (Thermo Fisher, Waltham, MA, USA). The UV–VIS diffuse reflectance spectrum was obtained using a UV–VIS spectrophotometer T9, Puxi, Beijing, with BaSO_4_ as the reference. The excitation spectrum and emission spectrum were measured using an RF-6000 spectral fluorescence photometer (Shimadzu Corporation, Kyoto, Japan), which used a xenon lamp as the excitation source. The ML spectrum was acquired from a custom-built uniaxial tensile testing machine for analysis using a fluorescence spectrophotometer (Omniλ300i, Zolix Instruments Co., Ltd., Zolix, Beijing, China). All measurements were performed at room temperature.

## 3. Results and Discussion

### 3.1. Phase and Crystal Structure Identification

Powder X-ray diffraction was utilized to analyze the crystal structure of materials with varying doping concentrations of Tb^3+^ and Sm^3+^. Figure 1a displays the XRD patterns of Ca1-xZnOS:xTb^3+^ (x = 0.01, 0.02, 0.03, 0.04, and 0.05), while Figure 1b shows the XRD pattern of Ca0.97-yZnOS:0.03Tb^3+^, ySm^3+^ (y = 0.0005, 0.001, 0.002, 0.005, and 0.01). By comparison, the characteristic diffraction peaks of all of the samples were accurately the same as those in the CaZnOS standard cards (PDF #01-072-3547) [2,23]. This indicated that the CaZnOS had been successfully prepared.

Figure 2a shows the Rietveld refinement pattern of the Ca0.96ZnOS:0.03Tb^3+^, 0.01Sm^3+^ sample. We performed Rietveld refinement on the XRD data of the Ca0.96ZnOS:0.03Tb^3+^, 0.01Sm^3+^ to analyze its crystal structure and sites. The experimental results agreed well with the calculated results. The reliability parameters of the Rietveld refinement are Rp = 9.46%, Rwp = 12.44%, and χ2 = 2.43. The cell parameters are a = 3.7608 Å, b = 3.7608 Å, and c = 11.3680 Å. The space group of CaZnOS is P63mc (186) with the crystal structure belonging to the hexagonal system [24,25]. Figure 2b illustrates the typical crystal structure of CaZnOS. The three-dimensional layered structure of CaZnOS is composed of a tetrahedron (ZnS_3_O) and an octahedron (CaO_3_S_3_). The coordination environment of the Zn atom is a tetrahedron consisting of three S atoms and one O atom, and the tetrahedrons are all arranged in parallel, forming a polar structure. The CaO_3_S_3_ octahedron is composed of a Ca atom surrounded by three S atoms and three O atoms [26]. A comparison of the ionic radii of Ca^2+^ (CN = 6, R = 1.00 Å), Zn^2+^ (CN = 4, R = 0.60 Å), Tb^3+^ (CN = 6, R = 0.92 Å), and Sm^3+^ (CN = 6, R = 0.96 Å) indicated that the ionic radii of Tb^3+^ and Sm^3+^ are similar to that of Ca^2+^, which means that Tb^3+^ and Sm^3+^ are more likely to occupy the lattice position of Ca^2+^ [27,28,29]. Figure 2c,d show that the unit cell parameters become smaller with increasing Sm content, indicating that Sm^3+^ (0.96 Å for CN = 6) replaces Ca^2+^ (1.00 Å for CN = 6).

Figure 3a presents an SEM image of the Ca0.968ZnOS:0.03Tb^3+^, 0.002Sm^3+^ sample. The sample is composed of small grains with a size smaller than 20 μm, which uniformly distribute together. The particle size of the sample concentrates at about 2–6 μm. Figure 3b shows the TEM image of the sample, which reveals a measured interplanar distance of 0.25 nm and corresponds to the (0 1 3) face of CaZnOS [30]. Figure 3c,d shows the energy-dispersive spectroscopy (EDS) elemental mapping images of the Ca0.968ZnOS:0.03Tb^3+^, 0.002Sm^3+^ sample. It can be seen that the Ca, Zn, O, S, Tb, and Sm atoms are evenly distributed throughout the sample, which is in line with the reaction reagent used in this experiment.

To further analyze the elemental distribution within the material, an X-ray energy spectrometer was utilized to test the PL material. Figure 4a depicts the XPS spectrum of the sample, confirming its composition of Ca, Zn, O, S, Tb, and Sm. Figure 4b–f sequentially display the XPS spectra of Ca, Zn, O, S, and Tb. The characteristic 2p3 and 2p1 signs of Ca appear at about 347 eV and 350 eV, respectively. Appearing at about 1022 eV and 1043 eV are the characteristic 2p3 and 2p1 signals of Zn. The characteristic 1s sign of O, the characteristic 2p sign of S, and the characteristic 4d sign of Tb appear at about 531 eV, 161 eV, and 161 eV, respectively [31,32].

### 3.2. Diffuse Reflection Spectrum and Band Gap Calculation

Figure 5a shows the diffuse reflectance spectrum of the Ca0.968ZnOS:0.03Tb^3+^, 0.002Sm^3+^ sample. As illustrated in Figure 5a, the diffuse reflection spectrum of the Ca0.968ZnOS:0.03Tb^3+^, 0.002Sm^3+^ sample exhibits a broad and intense absorption band at approximately 230~330 nm, which arises from the valence-to-conduction band transition within the principal lattice of CaZnOS [33]. Based on the reflection spectrum, we calculated the absorption spectrum of CaZnOS with the use of the Kubelka–Munk function [34]. Figure 5b illustrates the [F(R∞)hv]2 vs. photon energy (eV) plot for direct transitions of the Ca0.968ZnOS:0.03Tb^3+^, 0.002Sm^3+^ sample. Kubelka–Munk theory was used to estimate the band gap of Ca0.968ZnOS:0.03Tb^3+^, 0.002Sm^3+^ through the following equation:.
[F(R∞hv)]n = A(hv − Eg) (1)
where A represents a proportional constant, h represents the Planck constant, v represents the frequency, and Eg represents the value of the band gap. The nature of the sample decides the value of n; n = 2 for a direct transition [35]. In addition, F(R∞) is a Kubelka–Munk function defined as:F(R∞) = (1 − R∞)2/2R∞ = K/S (2)
where R represents the reflection, K represents the absorption, and S represents the scattering coefficient [36]. By extrapolating the linear part of the graphics to the axis of the abscissa, the Eg value can be estimated to be about 3.926 eV.

### 3.3. Excitation Spectrum and Emission Spectrum

Figure 6a shows the excitation spectra of the Ca1-xZnOS:xTb^3+^ (x = 0.01, 0.02, 0.03, 0.04, and 0.05) samples, which were monitored at 542 nm (the characteristic emission of Tb^3+^). The sample exhibits intense absorption at 254 nm, primarily attributed to the interband transition of the matrix material [37]. Figure 6b presents the emission spectra of the CaZnOS:Tb^3+^ sample under a 254 nm excitation light source. Due to the rich energy levels introduced by Tb^3+^ in the structure, CaZnOS:Tb^3+^ demonstrates excellent photoluminescence performance [38]. The typical emission peaks of Tb^3+^ appear at 385 nm, 418 nm, 439 nm, 494 nm, 548 nm, 586 nm, and 621 nm, respectively, corresponding to 5D3→7F6, 5D3→7F5, 5D3→7F4, 5D4→7F6, 5D4→7F5, 5D4→7F4, and 5D4→7F3 transitions of Tb^3+^. Due to cross-relaxation processes, the 5D3-level emission bands decrease with the concentration of Tb^3+^ [39]. It was observed that the PL intensity of the sample increased with the enhancement in Tb^3+^ and reached its maximum when the Tb^3+^ doping concentration was 0.03. Thereafter, the PL intensity of the sample decreased with the increase in Tb^3+^. The phenomenon can be explained by the fact that before reaching the critical concentration, the more Tb^3+^ in the luminous center, the better the PL performance of the sample. When the Tb^3+^ doping concentration reaches 0.03, the sample obtains the largest number of luminous centers and exhibits the best PL performance. However, when the Tb^3+^ doping concentration exceeds 0.03, instead of a further increase in PL intensity, there is a continuous decrease because of the concentration quenching. Therefore, it is concluded that Ca0.97ZnOS:0.03Tb^3+^ exhibits the best PL performance among all of the samples tested.

Figure 7 shows the emission spectrum of the Ca0.97-yZnOS:0.03Tb^3+^, ySm^3+^ (y = 0.0005, 0.001, 0.002, 0.005, and 0.01) under a 254 nm excitation light source [40]. Due to the doping of Tb^3+^ and Sm^3+^, the samples display excellent PL performance when excited. The typical emission peaks of Sm^3+^ appear at 552 nm, 628 nm, 682 nm, and 740 nm, respectively, corresponding to 4G5/2→6H5/2, 6H7/2, 6H9/2, and 6H11/2 transitions of Sm^3+^ [41]. With the increase in Sm^3+^ concentrations, the typical emission peaks of Sm^3+^ become more and more apparent.

Figure 8a displays the ML spectrum of the Ca1-xZnOS:xTb^3+^/PDMS (x = 0.01, 0.02, 0.03, 0.04, and 0.05) composite elastomer in the same condition. The phosphor/PDMS composite elastomer exhibits intense green mechanoluminescence when scratched. It is clear that the emission peaks are similar to their PL behaviors in Figure 6b, which belong to the characteristic radiative transfers of Tb^3+^ (5D4→7F6, 7F5, 7F4, and 7F3) [42]. CIE color coordinate calculation software was used to verify that the ML the Ca1-xZnOS:xTb^3+^/PDMS (x = 0.01, 0.02, 0.03, 0.04, and 0.05) composite elastomers are all green [43]. Figure 8b shows the CIE chromaticity diagram for the Ca1-xZnOS:xTb^3+^/PDMS composite elastomer at different Tb^3+^ doping concentrations. The CIE coordinates (x, y) of Ca0.97ZnOS:0.03Tb^3+^ are (0.3095, 0.4808).

Figure 9a shows the ML spectra of the Ca0.97-yZnOS:0.03Tb^3+^,ySm^3+^/PDMS (y = 0.0005, 0.001, 0.002, 0.005, and 0.01) composite elastomer under a stress of 10 N. It is evident that as the amount of Sm^3+^ increases, the characteristic emission peaks of Sm^3+^ become more prominent [44]. Similar to their PL behaviors shown in Figure 7, the phosphor/PDMS composite elastomer exhibits an intense green, yellow-green, and orange-red mechanoluminescence, corresponding to the characteristic radiative transfers of Sm^3+^ (4G5/2→6H5/2, 6H7/2, 6H9/2, and 6H11/2) [41]. Figure 9b shows the actual shooting pictures of the ML color, which correspond to the Sm^3+^ concentrations of 0, 0.0005, 0.001, 0.002, 0.005, and 0.01 in turn. The figure indicates that as the concentration of Sm^3+^ increases, the ML color of the PDMS composite elastomer shifts from green to yellow-green and finally transitions to orange-red. This demonstrates the ability to regulate the ML color of the composite elastomer from green light to orange-red light within a single substrate, which offers potential methods for future research. To research the influence of Sm^3+^ doping concentrations on the ML color of the sample, CIE color coordinate calculation software was utilized to analyze the sample. Figure 9c shows that the ML chromaticity points are located between the green and red regions. With the increase in Sm^3+^, the CIE coordinates (x, y) systematically change from (0.3167, 0.4751) to (0.4864, 0.4330). Figure 9d presents the CIE coordinates of different y values.

The CaZnOS:0.03Tb^3+^, ySm^3+^/PDMS (y = 0, 0.0005, 0.001, 0.002, 0.005, and 0.01) composite exhibited perfect ML performance and realized multicolor luminescence in the same substrate. Based on this, we designed an application for visual mechanical sensing in handwriting identification in Figure 10. The specific process is as follows: First, the phosphor samples measured by stoichiometry should be sintered by a solid-state method at a high temperature under an argon gas atmosphere. Second, after sintering, phosphor samples should be ground uniformly and then thoroughly mixed with PDMS and a curing agent. Thirdly, pour them into the mold and place them in the vacuum oven to dry. After drying, take out the phosphor/PDMS composite. Obvious mechanoluminescence was shown on the phosphor/PDMS composite when it was scratched with a sharp stick. Taking advantage of the ML performance, we scratched the phosphor/PDMS composite, and, at the same time, we took a time-lapsed shot to record the writing. The images of the ML intensity were obtained by using specific software to transform the shot data. The colors blue to red represent the ML intensity from weak to strong. For example, the stress of the written letter “O” at the beginning is significantly greater than the stress at the middle and end. Due to different people having different writing habits, the phosphor/PDMS composite exhibits different ML performance when used by different people. In the phosphor/PDMS composite, we realized multicolor luminescence in the same substrate. The phosphor/PDMS composite has the characteristics of simple preparation, strong ML intensity, and being reusable. The phosphor/PDMS composite has potential application value in handwriting identification, which also provides new ideas of mechanical sensing in terms of visual mechanics.

## 4. Conclusions

In conclusion, CaZnOS phosphors co-doped with Tb^3+^ and Sm^3+^ were successfully synthesized through a solid-state method at high temperature under an argon gas atmosphere, which had multicolor tunability and had no need for pre-irradiation. The XRD and Rietveld refinement confirmed that the co-doped Tb^3+^ and Sm^3+^ had no influence on the crystal structure of the CaZnOS and Tb^3+^ and Sm^3+^ ions that occupied the positions of Ca^2+^. SEM images described that the sample was composed of numerous small grains. TEM and elemental mapping characterizations were recorded simultaneously, which demonstrated that target elements and phases were distributed homogeneously throughout the sample. DRS images indicated that the valence-to-conduction band transition within the principal lattice of CaZnOS contributed to the absorption band in them. When excited at 254 nm, not only did the sample possess the green emission band at 545 nm derived from Tb^3+^ but also the orange-red emission bands at 552 nm, 628 nm, 682 nm, and 740 nm, derived from Sm^3+^ transitions. The ML spectra and CIE diagram showed that the ML color shifted from green to yellow-green and finally transitioned to orange-red, which was detected with the PDMS composite elastomer scratching. There were excellent ML performance and multicolor tunability in the CaZnOS:Tb^3+^, Sm^3+^ phosphor. The application applied in this work provides a novel strategy in visual mechanical sensing, which can facilitate further research into synthesizing novel ML phosphors and expand the promising applications of ML materials into more areas, especially visual mechanical sensing.

## Figures and Tables

**Figure 1 nanomaterials-14-01279-f001:**
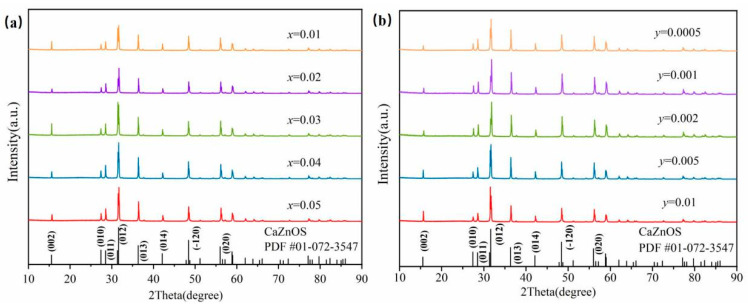
(**a**) XRD patterns of the Ca1-xZnOS:xTb^3+^ (x = 0.01, 0.02, 0.03, 0.04, and 0.05) samples; (**b**) XRD patterns of the Ca0.97-yZnOS:0.03Tb^3+^, ySm^3+^ (y = 0.0005, 0.001, 0.002, 0.005, and 0.01) samples.

**Figure 2 nanomaterials-14-01279-f002:**
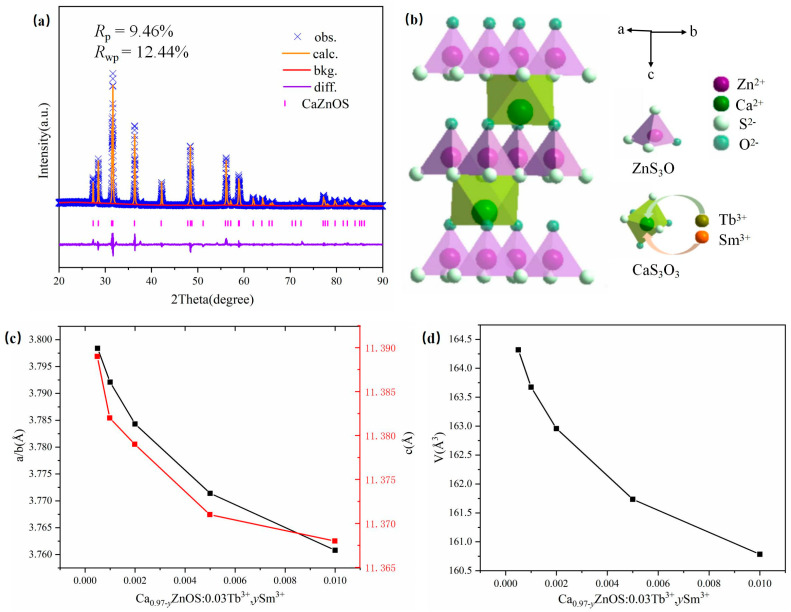
(**a**) The Rietveld refinement pattern of the Ca0.96ZnOS:0.03Tb^3+^0.01Sm^3+^ sample; (**b**) crystal structure of CaZnOS and schematic diagram of the Tb^3+^ and Sm^3+^ substitution process. (**c**) Variation in unit cell parameters (a, b, and c V) of Ca0.97-yZnOS:0.03Tb^3+^, ySm^3+^ solid-solution series dependent on y values. (**d**) Variation in unit cell parameters (V) of Ca0.97-yZnOS:0.03Tb^3+^, ySm^3+^ solid-solution series dependent on y values.

**Figure 3 nanomaterials-14-01279-f003:**
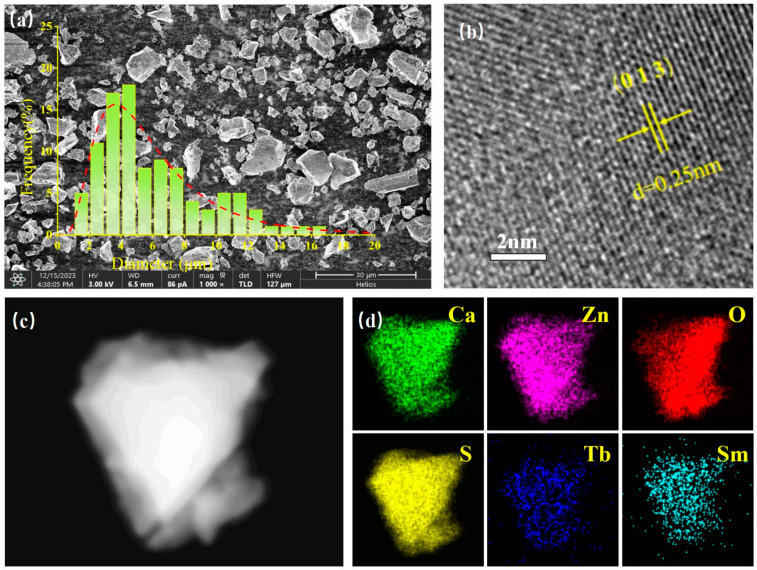
(**a**) SEM image of Ca0.968ZnOS:0.03Tb^3+^, 0.002Sm^3+^ samples (with sample particle size distribution map); (**b**) TEM image of the (0 1 3) crystal face; (**c**) the actual scanning area of the sample; (**d**) the energy-dispersive spectroscopy (EDS) elemental mapping images of the Ca0.968ZnOS:0.03Tb^3+^, 0.002Sm^3+^ material.

**Figure 4 nanomaterials-14-01279-f004:**
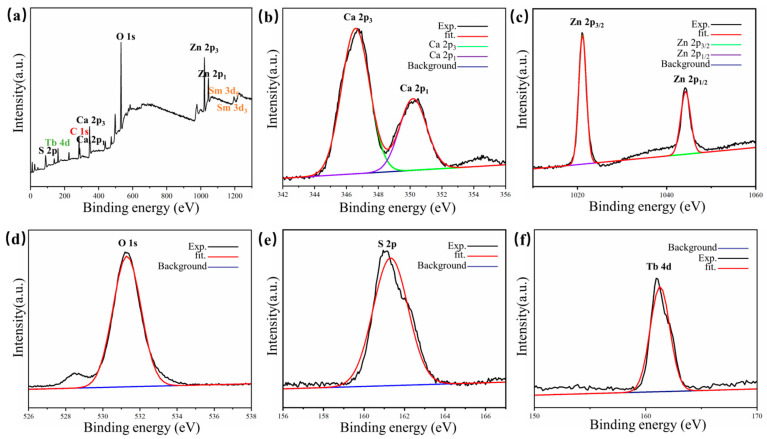
(**a**) XPS spectrum of Ca0.968ZnOS:0.03Tb^3+^, 0.002Sm^3+^ material; (**b**–**f**) high-resolution XPS spectra of Ca 2p, Zn 2p, O 1s, S 2p, Tb 4d in sequence.

**Figure 5 nanomaterials-14-01279-f005:**
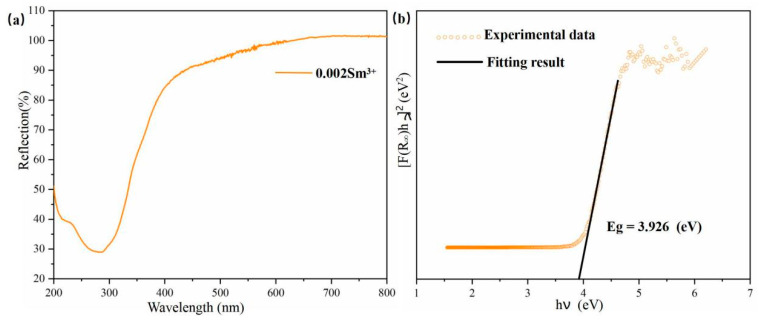
(**a**) Diffuse reflectance spectrum of the Ca0.968ZnOS:0.03Tb^3+^, 0.002Sm^3+^ sample; (**b**) the relationship between [F(R∞)hv]2 and photon energy Eg in Ca0.968ZnOS:0.03Tb^3+^, 0.002Sm^3+^.

**Figure 6 nanomaterials-14-01279-f006:**
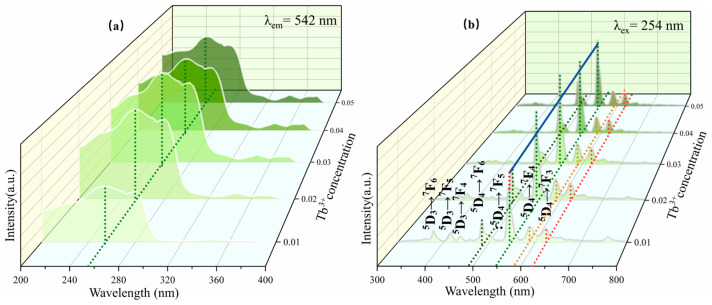
(**a**) Excitation spectra (λex = 542 nm) of Ca1-xZnOS:xTb^3+^ samples; (**b**) emission spectra (λem = 254 nm) of Ca1-xZnOS:xTb^3+^ samples (x = 0.01, 0.02, 0.03, 0.04, and 0.05).

**Figure 7 nanomaterials-14-01279-f007:**
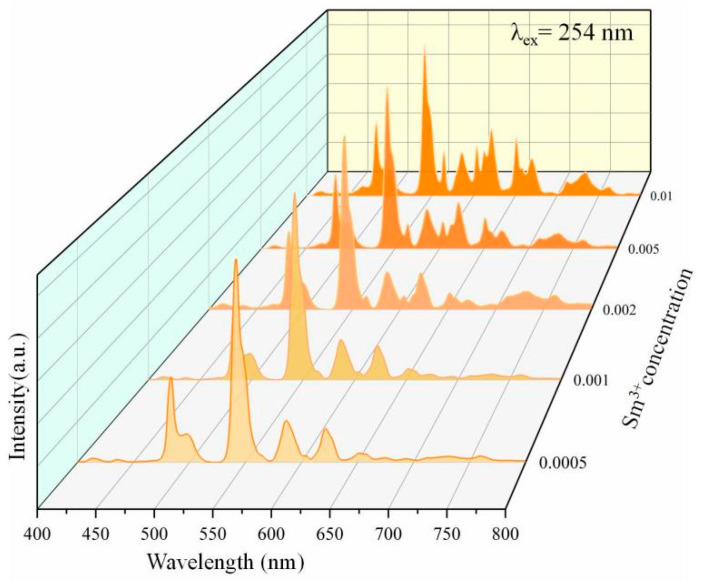
Emission spectrum of Ca0.97-yZnOS:0.03Tb^3+^, ySm^3+^ (y = 0.0005, 0.001, 0.002, 0.005, and 0.01).

**Figure 8 nanomaterials-14-01279-f008:**
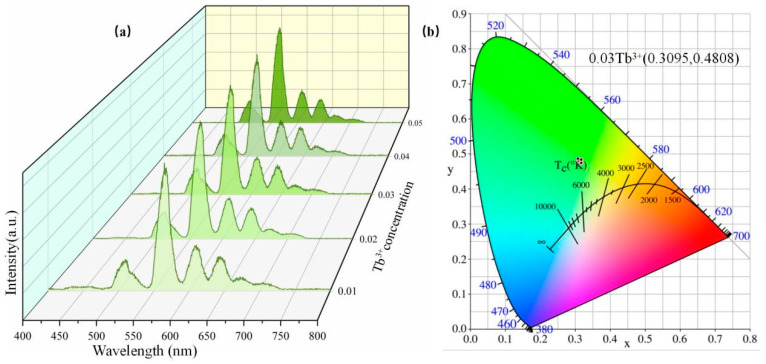
(**a**) The ML intensity image of the Ca1-xZnOS:xTb^3+^/PDMS composite elastomer at different Tb^3+^ doping concentrations; (**b**) CIE chromaticity diagram for Ca1-xZnOS:xTb^3+^ (x = 0.01, 0.02, 0.03, 0.04, and 0.05).

**Figure 9 nanomaterials-14-01279-f009:**
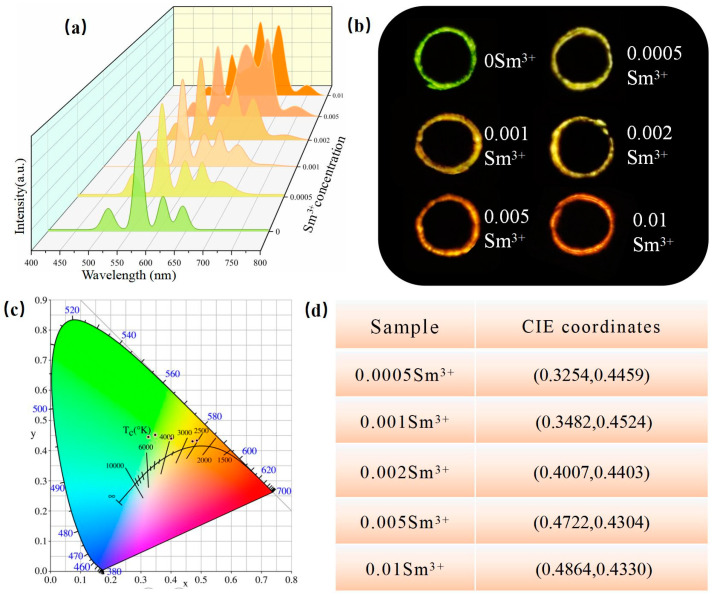
(**a**) The ML intensity diagram of the Ca0.97-yZnOS:0.03Tb^3+^,ySm^3+^/PDMS composite elastomer (y = 0.0005, 0.001, 0.002, 0.005, and 0.01); (**b**) the actual shooting pictures of the composite elastomer; (**c**) CIE chromaticity diagram for the Ca0.97-yZnOS:0.03Tb^3+^, ySm^3+^/PDMS composite elastomer; (**d**) the CIE coordinates of the samples.

**Figure 10 nanomaterials-14-01279-f010:**
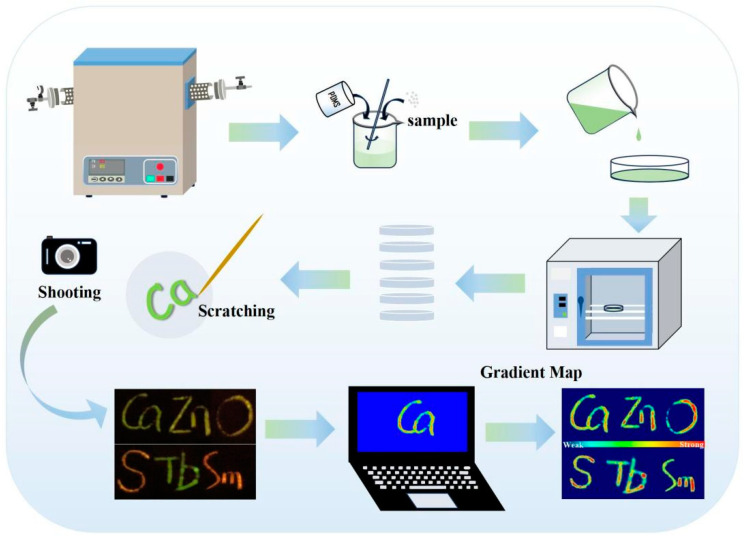
Demonstration of the application of visual mechanical sensing in handwriting identification.

## Data Availability

The data that support the findings of this study are available from the corresponding author upon reasonable request.

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
