# Peer review of "Achieving Tunable Mechanoluminescence in CaZnOS:Tb3+, Sm3+ for Multicolor Stress Sensing"

_nanomaterials, 2024, doi:10.3390/nano14151279_

Round 1

Reviewer 1 Report

Comments and Suggestions for Authors

The manuscript represents the development of a technique for obtaining a composite material based on CaZnOS:Tb3+,Sm3+ for stress sensing by means of mechanoluminescence. The paper topic is completely covering the scope of the Nanomaterials, paper conclusions are certainly interesting for the readership of the journal. The paper is structured, the material is presented in a consistent way, most conclusions are justified and supported by the results, results are presented in detail and accurately, and the English language is appropriate and understandable. The authors synthesized a series of solid solutions of  Ca1-xLnxZnOS (Ln=Sm3+ and Tb3+) composition and encapsulated them in polydimethylsiloxane matrix. The resulting materials demonstrated tunable mechanoluminescent properties that can be used for people's handwriting. The work provides an advance towards the visual mechanical sensing, which makes it publishable in the Nanomaterials journal upon some corrections.

1. In the introduction it is necessary to indicate what criteria must be fulfilled for a compound to possess mechanoluminescence. What exactly makes CaZnOS doped with rare-earth stand out among other mechanoluminescent compounds? What is the advantage of using rare-earth compounds in mechanoluminescence?

2. What is the novelty of the work? It should be highlighted in the abstract.

3. On page 2, line 72 perhaps an error, instead of CaCO it should be CaCO3? On page 2, line 72 please replace "Polydimethylsiloxane" with "polydimethylsiloxane".

4. In Section 3.1, the authors conclude that the CaZnOS phase is pure on the basis of XRD data, which is not quite correct, since the XRD method usually does not allow detection of impurities with a mass fraction of less than 5%.

5. It is necessary to clarify the lattice parameters for all obtained samples and find out how they change with the dopant content. The formation of a continuous series of solid solutions should be evidenced by a linear change in volume (or cubic root of volume) with the dopant content.

6. What is the purpose of measuring diffuse reflectance spectra and calculating the bandgap for the obtained samples? These parameters are somehow related to mechanoluminescence? Why are measurements made only for one sample of a specified composition?

7. Why no intrinsic absorption bands of rare-earth cations are observed in diffuse reflectance spectra?

8. What are the UV bands in the diffuse reflectance and luminescence excitation spectra related to? Why do the edges of their bands not match?

9. There is no description of the EDS method in the experimental part of the paper.

10. On page 6, line 186, please replace "Photoluminescence" with "photoluminescence".

Author Response

Comment 1: In the introduction it is necessary to indicate what criteria must be fulfilled for a compound to possess mechanoluminescence. What exactly makes CaZnOS doped with rare-earth stand out among other mechanoluminescent compounds? What is the advantage of using rare-earth compounds in mechanoluminescence? 

Response 1: Thank you for your good suggestion. I have revised them according to your suggestion and all the revisions have been labeled by red color in the manuscript. On page 1, line 56 has been revised to "CaZnOS was chosen as the host materials in our study owing to its convenience for impurity doping and band gap for transition." The CaZnOS host contains two types of cation sites, namely Ca2+ (CN = 6, R = 1.00 Å) and Zn2+ (CN = 4, R = 0.60 Å), which are able to accommodate transition lanthanides. The band gap of CaZnOS is about 3.926 eV, making it a better semiconductor as a promising host lattice for activators.[1] Rare earth because of its special electron layer structure, and has the spectral properties of the general elements can not be matched, rare earth luminescence almost covers the whole category of solid luminescence. The atoms of rare earth elements have an unfilled 4f, 5d electron configuration shielded by the outside world, so there are rich electronic energy levels and long-lived excited states, and more than 200,000 energy level transition channels can produce a variety of radiation absorption and emission, constituting a wide range of luminous materials.

  • Du,Y Jiang, Y.; Sun, T.; Zhao, J.; Huang, B.; Peng, D.; Wang, Feng. Mechanically Excited Multicolor Luminescence in Lanthanide I Advanced Materials 2019, 31 (7), 1807062.

Comment 2: What is the novelty of the work? It should be highlighted in the abstract. 

Response 2: Thank you for your good suggestion. I have revised them according to your suggestion and all the revisions have been labeled by red color in the manuscript. On page 1, line 24 has been revised to "Taking a time-lapse shoot while writing, we can easily obtained the images of the writer’s handwriting. The images of the ML intensity were acquired by using the specific software to transform the shooting data. We can easily distinguish people's handwriting through analyzing the different ML performance." 

Comment 3: On page 2, line 72 perhaps an error, instead of CaCO it should be CaCO3? On page 2, line 72 please replace "Polydimethylsiloxane" with "polydimethylsiloxane".

Response 3: Thank you very much for pointing out my mistakes. This error has been corrected in the manuscript and the revision has been labeled by red color. On page 2, line 72 has been revised to "Raw materials of the sample are CaCO3 (A.R.), ZnS (A.R.), Tb4O7 (A.R.), Sm2O(A.R.), and Li2CO3 (A.R.)." On page 2, line 81 has been revised to "The polydimethylsiloxane (PDMS) served as an elastic matrix, which offered internal stress to the ML phosphor."

Comment 4: In Section 3.1, the authors conclude that the CaZnOS phase is pure on the basis of XRD data, which is not quite correct, since the XRD method usually does not allow detection of impurities with a mass fraction of less than 5%.

Response 4: Thank you very much for your useful comments.This mistake is due to my incorrect expression. We are sorry that didn't express this issue correctly. And this revision has been labeled by red color. On page 3, from line 109 to line 111, the conclusion has been revised to "This indicated that CaZnOS had been successfully prepared."

Comment 5: It is necessary to clarify the lattice parameters for all obtained samples and find out how they change with the dopant content. The formation of a continuous series of solid solutions should be evidenced by a linear change in volume (or cubic root of volume) with the dopant content. 

Response 5: We gratefully appreciate for your valuable suggestion. We have added Figure. (c) and Figure. (d), and the descriptions corresponding to the figures have also been added in the manuscript as described below:

From Figure. (c) and Figure. (d), it is clear that with the increase of Sm, the radii have become smaller, which further indicates that Sm ions replaces Ca ions.

Comment 6: What is the purpose of measuring diffuse reflectance spectra and calculating the bandgap for the obtained samples? These parameters are somehow related to mechanoluminescence? Why are measurements made only for one sample of a specified composition?

Response 6: Thank you very much for your constructive comments. The diffuse reflectance spectra were measured to prove that the band gap of CaZnOS is about 3.926 eV, which can accommodate transition lanthanides and is a better semiconductor as a promising host lattice for activators. According to your suggestions, we measured a series of CaZnOS:Tb3+,Sm3+ phosphors and conducted a new analysis on this part of the study. The figures below are the diffuse reflectance spectrum of Ca1-x-yZnOS:xTb3+,ySm3+ (x = 0.01, 0.02, 0.03, 0.04 and 0.05, when x = 0.03, y = 0.0005, 0.001, 0.002, 0.005 and 0.01) samples. As illustrated in the figures, the diffuse reflection spectrum of the samples similarly exhibits a broad and intense absorption band at approximately 230 ~ 330 nm. Since the absorption bands are similar, we chose the Ca0.968ZnOS:0.03Tb3+,0.002Sm3+ sample to focus on the analysis.

Comment 7: Why no intrinsic absorption bands of rare-earth cations are observed in diffuse reflectance spectra?

Response 7: Thank you very much for your careful review and constructive comments concerning our manuscript. There are intrinsic absorption bands of rare-earth cations in the diffuse reflectance spectra, but it was too weak to be seen. Therefore, we focus on the valence-to-conduction band transition within the principal lattice of CaZnOS. 

Comment 8: What are the UV bands in the diffuse reflectance and luminescence excitation spectra related to? Why do the edges of their bands not match?

Response 8: Thanks very much for your good suggestion. I am sorry that the manuscript we submitted before did not carry out a detailed and professional study on this part of the work. We measured the photoluminescence spectra of the samples under a 254 nm excitation light source. The energy absorbed by the light stimulation is all used for light. In tests of diffuse reflectance spectroscopy, the energy absorbed by the material is only partially used for light, and there are also other forms of energy release, such as heat. The forms of forms are different, so the edges of their bands do not match. 

Comment 9: There is no description of the EDS method in the experimental part of the paper.

Response 9: We thank the reviewer for pointing out this issue. The mistake is due to my negligence. As shown in Line 60, I have added the method of EDS according to your suggestion in the manuscript. On page 2, line 57 has been revised to "A series of CaZnOS:Tb3+,Sm3+ phosphors were investigated by characterizations and performance tests in detail, such as X-ray diffraction (XRD), scanning electron microscopy (SEM), energy-dispersive spectroscopy (EDS), transmission electron microscopy (TEM), X-ray photoelectron spectroscopy (XPS), diffuse reflectance spectrum (DRS), excitation spectrum and emission spectrum." On page 2, line 92 has been revised to "Transmission electron microscopy images were captured with the FEI Tecnai F30 instrument. The morphology and elemental mapping of typical samples were characterized using Zeiss ZEISS-6035 scanning electron microscopy."

Comment 10: On page 6, line 186, please replace "Photoluminescence" with "photoluminescence".

Response 10: Thank you very much for pointing out our mistake. We have revised it according to your suggestion. On page 6, line 186 has been revised to "Figure. 6(b) presented the emission spectra of the CaZnOS:Tb3+ sample under a 254 nm excitation light source. Due to the rich energy levels introduced by Tb3+ in the structure, CaZnOS:Tb3+ demonstrates excellent photoluminescence performance."

Reviewer 2 Report

Comments and Suggestions for Authors

The authors shows an interesting paper based on mechano-luminiscence properties obtained CaZnOS:Tb3-Sm3+ materials. I consider this paper very interesting and specially the application of visual mechanical sense in handwriting identification. However I have several comments before accepting this paper:

Line 187. In the emission spectrum of Tb appear emission bands below 500 nm that are not identified by the authors. These bands correspond to emission from the 5D3 level and decrease with the concentration of Tb due to cross relaxation processes.

Line 200. There is a mistake. The excitation spectrum is obtained detecting at 542 nm and the emission spectrum is obtained exciting at 254 nm.  

Line 207. In the emission spectrum with high concentration of Sm appear sharp peaks. Do these peaks correspond to Sm emissions?

Line 209. Why under ML experiments the emission of Sm is very intense (Fig. 9) but under UV excitation it is not so evident (Fig. 7)?

Author Response

Comment 1: Line 187. In the emission spectrum of Tb appear emission bands below 500 nm that are not identified by the authors. These bands correspond to emission from the 5D3 level and decrease with the concentration of Tb due to cross relaxation processes. 

Response 1: Thank you very much for your constructive comments. According to your suggestion, we have added new data and conducted new analysis on this part of the study. Line 189 has been revised to "The typical emission peaks of Tb3+ appear at 385nm, 418nm, 439nm, 494 nm, 548 nm, 586 nm, and 621 nm, respectively corresponding to 5D3 → 7F6, 5D3 → 7F5, 5D3 → 7F4, 5D4 → 7F6, 5D4 → 7F5, 5D4 → 7F4, and 5D4 → 7F3 transitions of Tb3+.[1] Due to cross relaxation processes, the 5D3 level emission bands decrease with the concentration of Tb3+." Figure 6 (a) has been revised to the picture below.

[1] Fan, X.; Xu, X.; Yu, X.; Chen, W.; Zhou, D.; Qiu, J. Wide band long persistent luminescence of Ca3Ga2Ge3O12: Tb3+, Tm3+ phosphor with synergistic effect of different traps. Materials Research Bulletin 2018 398402.

Comment 2: Line 200. There is a mistake. The excitation spectrum is obtained detecting at 542 nm and the emission spectrum is obtained exciting at 254 nm.

Response 2: Thank you very much for pointing out our mistake. We deeply apologize for the incorrect expression. We worked on the manuscript for a long time and the repeated addition led to incorrect expression. This error has been corrected in the manuscript and the revision has been labeled by red color. And the figure has been revised in the manuscript. Line 200 has been revised to “Excitation spectra (λex = 542 nm) of Ca1-xZnOS:xTb3+ samples; (b) Emission spectra ((λem = 254 nm) of Ca1-xZnOS:xTb3+ samples (x = 0.01, 0.02, 0.03, 0.04, and 0.05).”

Comment 3: Line 207. In the emission spectrum with high concentration of Sm appear sharp peaks. Do these peaks correspond to Sm emissions?

Response 3: Thank you very much for your constructive comments. In the course of our research, we ignored this question. Previously, we reviewed relevant literature on Sm emissions, there are some researches show that Sm ions appear sharp peaks with high concentration. The peaks figure 7 showed are belonging to Sm.

Comment 4: Why under ML experiments the emission of Sm is very intense (Fig. 9) but under UV excitation it is not so evident (Fig. 7)?

Response 4: Thank you very much for your careful review and constructive comments. We measured the photoluminescence spectra of the samples under a 254 nm excitation light source. There were typical emission peaks of Tb3+ and Sm3+, which belong to light stimulation photoluminescence. Based on the emission spectra, we measured the excitation spectra of the samples. As can be seen by the excitation spectrum, the best excitation is not the same. ML spectrum was acquired from the custom-built uniaxial tensile testing machine for analysis by the fluorescence spectrophotometer. Type of this kind testing machine is used to apply stretch force to the sample and a fluorescence spectrophotometer is used to collect the photoluminescence signal. The principles of luminous are different, so the emissions of Sm is different.

Round 2

Reviewer 1 Report

Comments and Suggestions for Authors

The manuscript has been sufficiently finalized, but a few comments remain:
1. On page 4, lines 144-145, the sentence "From Figure. (c) and Figure. (d), it is clear that with the increase of Sm,the radii have become smaller, which further indicate that Sm ions replaces Ca ions." should be changed to "Figure. (c) and Fig. (d) show that the unit cell parameters become smaller with increasing Sm content, indicating that Sm3+ (0.96 Å for CN=6) replaces Ca2+ (1.00 Å for CN=6)".
2. In Figures 2c and c, the caption "Ca0.97-yZnOS:0.03Tb3+,ySm3+" should be changed to "y" or "Sm3+ content".

Author Response

Comment 1: On page 4, lines 144-145, the sentence "From Figure. (c) and Figure. (d), it is clear that with the increase of Sm,the radii have become smaller, which further indicate that Sm ions replaces Ca ions." should be changed to "Figure. (c) and Fig. (d) show that the unit cell parameters become smaller with increasing Sm content, indicating that Sm3+ (0.96 Å for CN=6) replaces Ca2+ (1.00 Å for CN=6)".

Reviewer 1: Thank you for your good suggestion. I have revised them according to your suggestion and all the revisions have been labeled by red color in the manuscript.
Comment 2: In Figures 2c and c, the caption "Ca0.97-yZnOS:0.03Tb3+,ySm3+" should be changed to "y" or "Sm3+ content".

Reviewer 2: Thank you very much for your careful review and constructive comments. Figures 2 (c) and (d) has been revised to the picture below.
